# Cannabidiol interactions with voltage-gated sodium channels

Lily Goodyer Sait[1†], Altin Sula[1†], Mohammad-Reza Ghovanloo[2†], David Hollingworth[1], Peter C Ruben[2], BA Wallace[1*]

[1]Institute of Structural and Molecular Biology, Birkbeck College, University of London, London, United Kingdom; [2]Department of Biomedical Physiology and Kinesiology, Simon Fraser University, Burnaby, Canada

**Abstract** Voltage-gated sodium channels are targets for a range of pharmaceutical drugs developed for the treatment of neurological diseases. Cannabidiol (CBD), the non-psychoactive compound isolated from cannabis plants, was recently approved for treatment of two types of epilepsy associated with sodium channel mutations. This study used high-resolution X-ray crystallography to demonstrate the detailed nature of the interactions between CBD and the NavMs voltage-gated sodium channel, and electrophysiology to show the functional effects of binding CBD to these channels. CBD binds at a novel site at the interface of the fenestrations and the central hydrophobic cavity of the channel. Binding at this site blocks the transmembrane-spanning sodium ion translocation pathway, providing a molecular mechanism for channel inhibition. Modelling studies suggest why the closely-related psychoactive compound tetrahydrocannabinol may not have the same effects on these channels. Finally, comparisons are made with the TRPV2 channel, also recently proposed as a target site for CBD. In summary, this study provides novel insight into a possible mechanism for CBD interactions with sodium channels.

*For correspondence:
b.wallace@mail.cryst.bbk.ac.uk

†These authors contributed equally to this work

Competing interests: The authors declare that no competing interests exist.

## Introduction

Voltage-gated sodium channels (Navs) specifically enable the passage of sodium ions across cell membranes, contributing to the electrical signalling in cells (*Ahern et al., 2016*). Nine homologous human sodium channel subtypes, designated hNav1.1-hNav1.9 have been identified, which have different functional characteristics and expression profiles within different tissues (*Catterall et al., 2005*). Mutations in hNavs have been associated with a range of channelopathies, including pain, epilepsy, and heart disorders, making them major targets for drug development (*Bagal et al., 2015*; *Kaplan et al., 2016*).

Cannabinoids (including cannabidiol (CBD) and Δ-9-tetrahydrocannabinol (THC)) are hydrophobic compounds produced by the cannabis plant. Whilst THC has been primarily associated with psycho-active drug use (*Rosenberg et al., 2015*; *Pisanti et al., 2017*), the non-psychoactive component, CBD, has been shown to have clinical applications as a therapeutic drug for treatment of epileptic conditions (*Rosenberg et al., 2017*), and was recently approved by the European Medicines Agency and the Federal Drug Administration for use in children for treatment of Dravet Syndrome and Lenox-Gastaut Syndrome (*Sarker and Nahar, 2020*). Both of these diseases are rare early-onset epilepsies associated with Navs, with Dravet patients often having mutations in the hNav1.1 gene *SCN1A* (*Marini et al., 2011*). Despite a significant amount of evidence reporting on the effectiveness of CBD for treating epileptic conditions (*Cross et al., 2017*; *Devinsky et al., 2018*), the molecular basis of its target interactions with ion channels and receptors still remains unclear (*Watkins, 2019*).

Functional studies have suggested that CBD both blocks the pore and stabilizes the inactivated states of sodium channels. It inhibits the channel activities of human Nav1.1-Nav1.7 isoforms, as well as those of the prokaryotic Nav homologue NaChBac, with $IC_{50}$s ranging from 1.5 to 3.8 μM,

suggesting inhibition occurs at physiologically-relevant concentrations (*Patel et al., 2016*; *Ghovanloo et al., 2018*). This preference for the inactivated state is a characteristic of classic pore blockers (*Kuo and Bean, 1994*). Furthermore the resurgent and persistent sodium currents of hNav1.2 have been shown to be inhibited at CBD concentrations of 1 µM (*Mason and Cummins, 2020*). However, to date, the structural basis of the interactions of CBD and Navs has not been identified on a molecular level.

In the present study, in order to examine the nature of the interactions of CBD with sodium channels, the high-resolution crystal structure of a complex of CBD with the NavMs voltage-gated sodium channel from *M. marinus* (*Sula et al., 2017*) was determined, enabling the binding sites for the CBD molecule to be clearly defined. The NavMs channel has been shown to be an excellent exemplar for hNavs as it exhibits highly similar functional (*Bagnéris et al., 2014*; *Ke et al., 2018*), conductance (*Ulmschneider et al., 2013*), and drug-binding (similar $IC_{50}$ values) characteristics (*Bagnéris et al., 2014*), as well as sequence and structural homologies (*Sula et al., 2017*; *Sula and Wallace, 2017*).

This high-resolution crystallographic study of a sodium channel-CBD complex, combined with functional studies of CBD on this channel, thus provides the means for both understanding the molecular interactions of CBD and Nav targets, and how these may be related to its use for treatment of epilepsy, and possibly other channelopathies.

## Results

This study utilised the prokaryotic NavMs voltage-gated sodium channel to examine the site of interactions of the naturally-occurring non-psychoactive CBD compound isolated from cannabis plants. NavMs channels are tetramers with each monomer consisting of six transmembrane (TM) helices (4 of which form each of the voltage sensor subdomains and 2 of which form the pore subdomains), whilst all of the hNav channel isoforms are monomers of four similar but not identical domains (each of which also consists of 6 transmembrane helices that are comprised of 4-helical voltage sensor subdomains and 2-helical pore subdomains) plus inter-domain loop regions (which differ considerably between hNavs, but are not involved in the CBD-binding sites).

The compelling reason for using crystal structures of NavMs in this study is that they provide the, to date, highest resolution (~2.2–2.5 Å) views of any sodium channel (*Naylor et al., 2016*; *Sula et al., 2017*), especially of the TM and drug-binding regions, thus enabling detailed views of the protein molecular structures with drugs bound to them. This, combined with its strong sequence, structural and functional similarities (*Sula et al., 2017*; *Sula and Wallace, 2017*) to hNav1.1 and hNav1.2 channels, enables important comparisons with sodium channels found in human brain tissues. Structure/function/drug-binding studies using some of the other prokaryotic sodium channels have also been used for drug discovery projects (*Martin and Corry, 2014*; *Ouyang et al., 2007*), but their structures tend to be of lower resolution than those of NavMs. Furthermore the cryo-EM structures of hNavs available to date generally have overall resolutions of ~4–5 Å, with the transmembrane regions having the best resolutions of ~3 Å (not sufficient for detailed views of the binding sites), whilst their extra- and intra- membranous regions are less well defined. However, these reasons would not be sufficient to indicate the value of using NavMs for understanding the molecular basis of drug interactions if its functional properties (conductance and drug-binding affinities) were not comparable to those of hNavs, but NavMs and hNav1.1 have been shown to exhibit similar ion flux and other conductance properties, as well as very similar binding affinities for a wide range of sodium channel-specific drugs (*Bagnéris et al., 2014*).

### The CBD-binding site in sodium channels

The CBD-binding site is located and clearly visible (*Figure 1* and *2*, *Figure 2—figure supplement 1*) in a well-defined region of the NavMs-CBD structure, sited in the hydrophobic pockets present between subunits that run perpendicular to the channel direction (*Montini et al., 2018*) (such features have been designated 'fenestrations' and are located (horizontally in *Figure 1*) in the transmembrane region, just below the level of the selectivity filter), and are the features originally proposed by *Hille, 1977* as sites for ingress of hydrophobic drugs into the channel interior. CBD is located at the ends of the fenestrations that lie closest to the central pore, and protrudes into (and blocks) the central transmembrane cavity, just below the sodium ion selectivity filter (*Figure 1C*). There is experimental electron density (*Figure 2A*, right panel) and enough room (*Figure 2A*, third

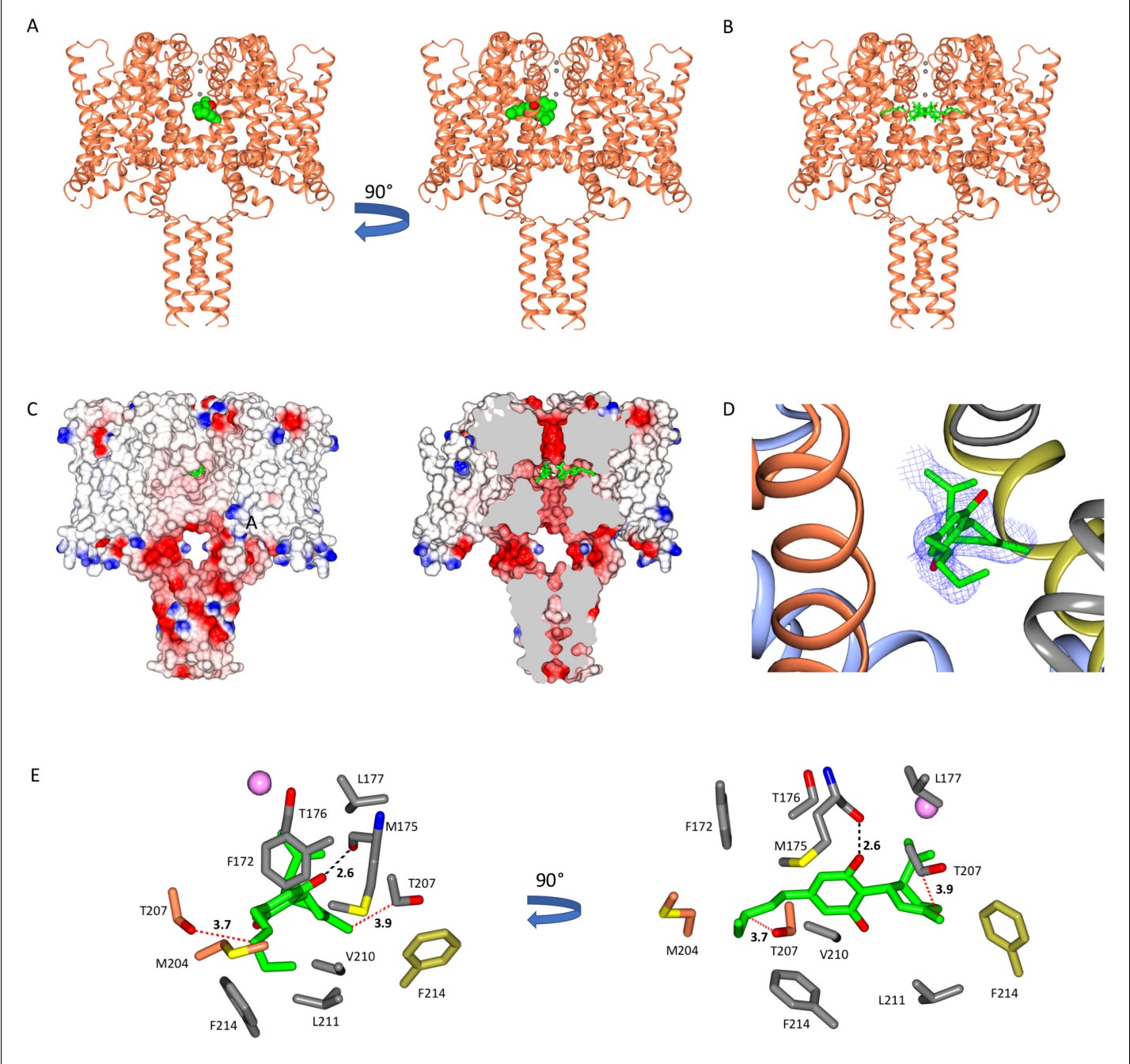

**Figure 1.** The NavMs sodium channel-cannabidiol (CBD) crystal structure. (**A**) The crystal structure (2.3 Å resolution) of the NavMs sodium channel (in coral coloured ribbon depiction) with one CBD molecule (in green space-filing depiction), showing its location within the hydrophobic cavity of the channel, located adjacent to the fenestration. Three sodium ions are shown as grey spheres in the selectivity filter, for visual reference. The view on the right is rotated 90 degrees from the view on the left. (**B**) As in (**A**) but showing 4 CBD molecules in stick depiction. (**C**) (left) Surface view of space filling structure of NavMs coloured by electrostatics with CBD (in green) present. The orientation is the same as in the left panel of part A. The CBD (in green) is only just visible through the exterior end of one of the fenestration holes. (right) As in panel B, but with a slice through the space filling model (and through the middle of the fenestrations). For clarity only 2 CBD molecules are included, showing where the drug lies along the fenestration and blocks the ion pathway. (**D**) The CBD-binding site: the polypeptide backbone of the NavMs-CBD complex is depicted in ribbon motif. The ribbons are coloured by subunit (regions of the subunits that come in close contact with the one CBD molecule shown, are depicted in red, grey, blue, and yellow). The (2Fo-Fc) map (in blue mesh) was calculated at 0.7 sigma and the structure of the CBD molecule present is in stick depiction. (**E**) Detailed views of the residues that lie within 3.9 Å of the CBD molecule (which is depicted in green/red stick representation) are shown and coloured by domain (as in part D). The H-bond between CBD and NavMs involves the M175 backbone carbonyl group, which is shown as a dashed black line. The distance between the side chain of T207 and the CBD molecule is 3.7 Å, and is indicated by a dotted red line. This is the residue that was mutated in the

electrophysiology studies. The pink sphere indicates the sodium ion site in the selectivity filter which is furthest from the extra-membranous surface (the one located farthest into the channel). On the right is the same view, rotated by 90 degrees.

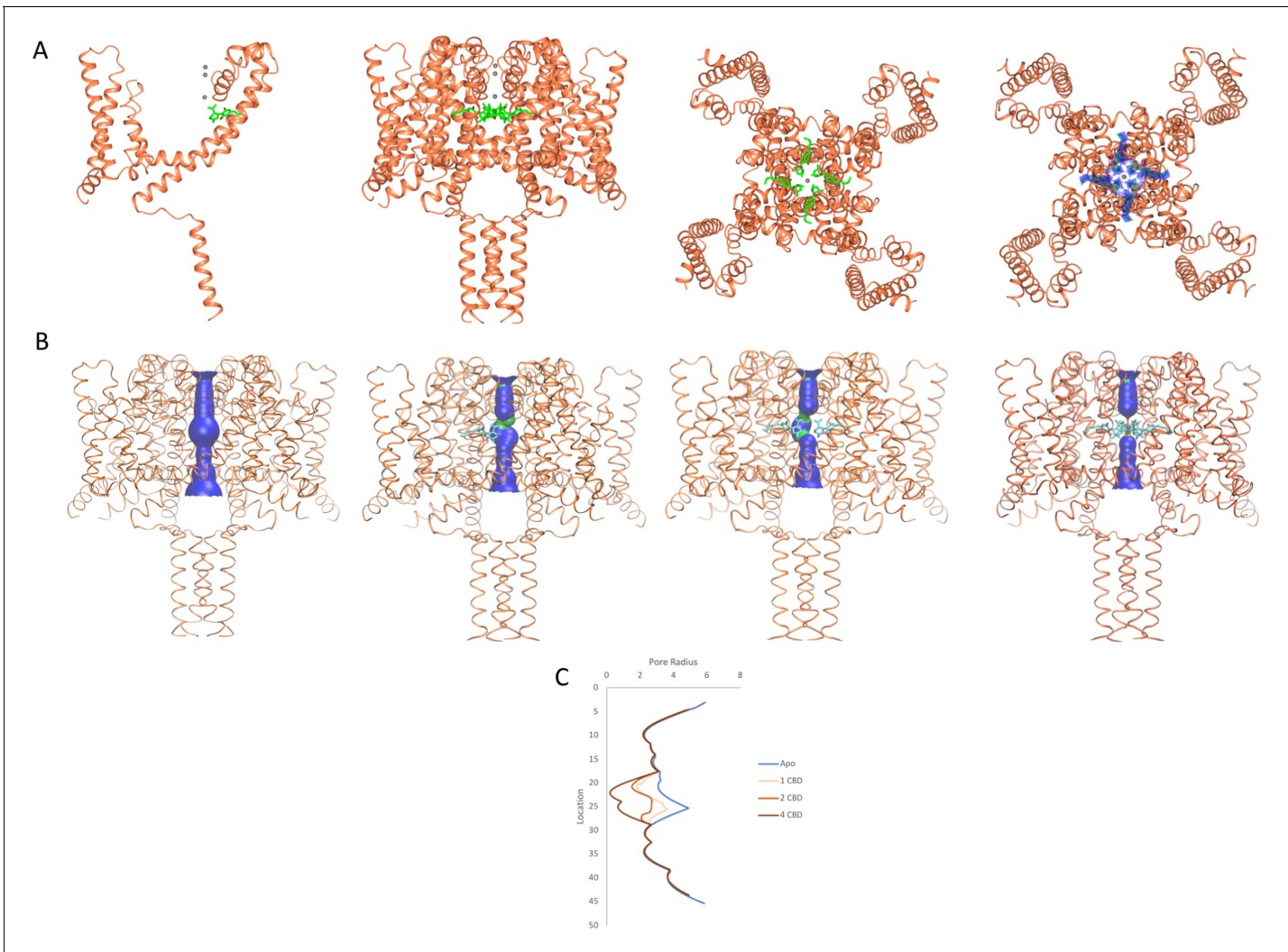

**Figure 2.** NavMs structures and pore diameters in the absence and presence of CBD. The NavMs structure (coral) is depicted in ribbon motif. (**A**) (from the left): For clarity, only a single NavMs monomer with one CDB (green sticks) is shown; the NavMs tetrameric structure with all 4 CBDs bound; top view of NavMs tetramer with all 4 CDBs bound; top view of NavMs tetramer showing the electron density map (blue) demonstrating all 4 CBDs are bound in the tetrameric structure. (**B**) Pore interior dimensions calculated using the HOLE algorithm (**Smart et al., 1993**). Progressively from left to right: The apo structure and structures with 1, 2, and 4 CBD molecules present. In this figure the HOLE surface is depicted in blue for pore radii greater than 2.3 Å, and green for radii less than 2.3 Å. There is no occlusion in the absence of CBD, so hydrated sodium ions could freely pass through the pore. The NavMs structure with 4 CBD molecules present shows a full occlusion in the middle of the channel transmembrane pathway (near the centre of the hydrophobic cavity, so ion transport would be prevented [**Naylor et al., 2016**]). (**C**) Accessibility plots of pore radii versus position within the pore, in the absence and presence of different numbers of CBD molecules. The plot for the apo structure is in blue, and the plots for the CBD-containing structures with 1, 2, or 4 CBD molecules are in coral, red and black, respectively. The plots for 2 and 3 CBD molecules were the same, so the latter is not shown. In cases where at least some region of the radius is <2.0 Å, sodium ions will not be able to be translocated across the channel (**Naylor et al., 2016**). This plot thus shows that regardless of whether one or more CBD molecules are present, ion passage will not occur. These figures were produced using VMD software (**Humphrey et al., 1996**).

The online version of this article includes the following figure supplement(s) for figure 2:

**Figure supplement 1.** Comparisons of the electron density map of CBD in the NavMs-CBD complex with the electron density maps of Apo-NavMs crystals.

panel from left) for four CBD molecules in this region, although one would be sufficient to block sodium ion passage, as seen from the HOLE (*Smart et al., 1993*) depictions (*Figure 2B*) and pore radius plots (*Figure 2C*), which show the size of the transmembrane pathway with and without different numbers of CBD molecules; their blockage clearly provides a mechanism for channel inhibition as well as a basis for understanding the concentration-dependence of the drug effects. Each binding site is comprised of 11 residues from three different subunits in NavMs (shown in detail and in different colours in *Figure 1D & E*). The corresponding residues in hNavs arise from three different domains of the same polypeptide chain (*Figure 3*). It should be noted here, that this region of the apo structure also exhibits some electron density (but has a different size and shape), which has been attributed to detergent molecules. The (2Fo-Fc) electron density map of the CBD complex (*Figure 2—figure supplement 1*) clearly indicates that in these crystals the site is occupied by CBD rather than by detergent.

The location of the CBD-binding site is very close to the locations of the binding sites that have been identified for analgesic and other hydrophobic compounds in NavMs (*Bagnéris et al., 2014*; *Figure 4*) as well as in another bacterial sodium channel, NavAb (*Gamal El-Din et al., 2018*). This is of interest because those other compounds also inhibit hNav functions, and so suggest the importance of this site for drug interactions in humans. All of the interactions seen except one, that of residue M175 (*Figure 1E*), involve hydrophobic interactions rather than hydrogen-bond formation (but that particular interaction between the main chain carbonyl of residue M175 and the OH group present in CBD, may be important for specificity of binding – see below).

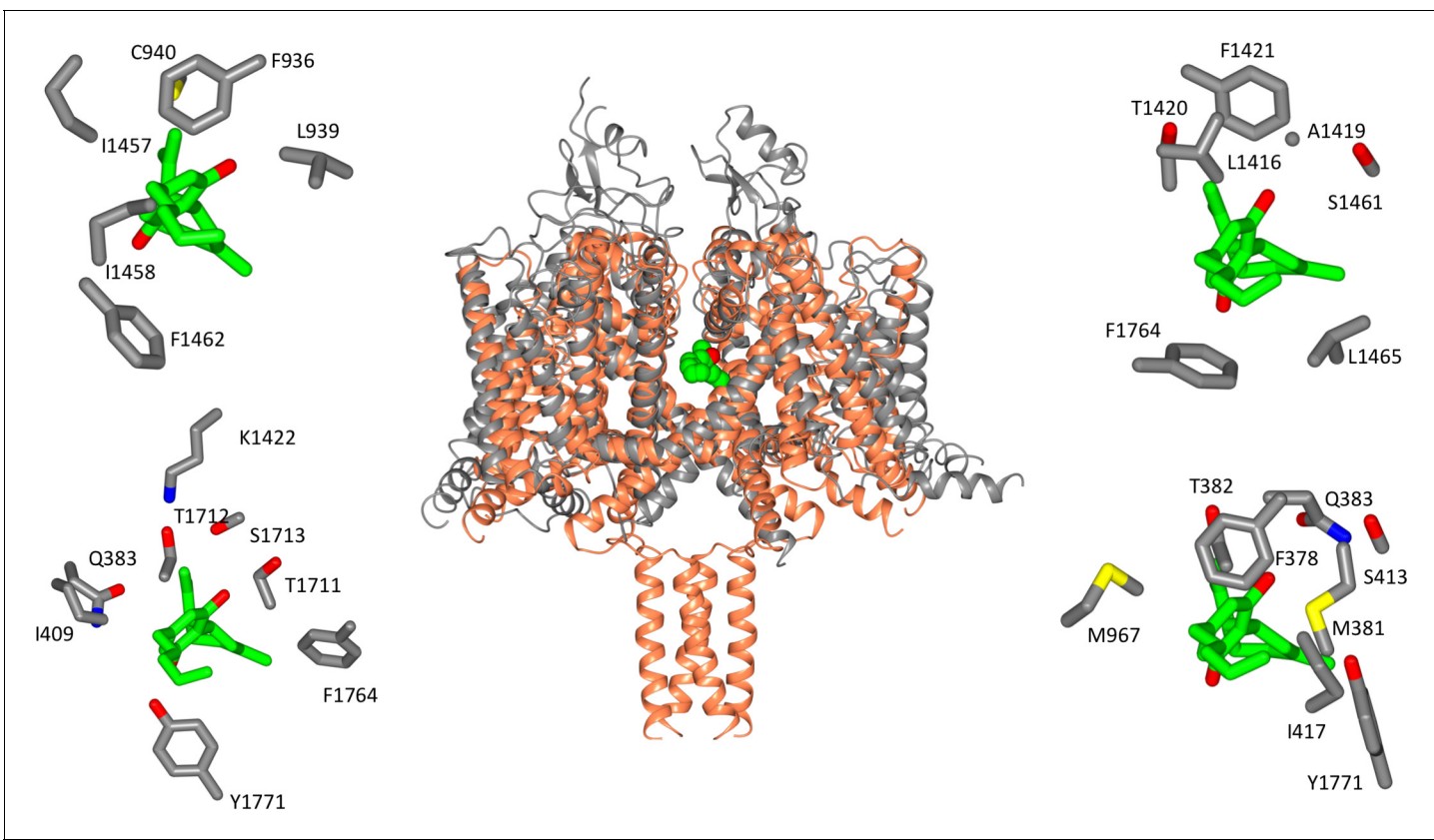

**Figure 3.** Location of CBD-binding sites in NavMs and the equivalent sites in hNav1.2. (Centre) Structural alignment of the NavMs-CBD crystal structure (coral) and the hNav1.2 cryo-EM structure (grey). The RMSD of the aligned structures is 3.2 Å. (Top left): equivalent binding residues between domain I and domain II of hNav1.2 found within 4 Å of the CBD site. (Top right): equivalent binding residues in hNav1.2 between domains II and III located within 4 Å of the CBD-binding site. (Bottom left): Residues in hNav1.2 between domains III and IV within 4 Å of the CBD-binding site. (Bottom right): Residues in hNAv1.2 between domains IV and I within 4 Å of the CBD-binding site. In the surrounding panels the atoms in the protein are coloured by atom type, with carbons represented in grey, oxygen in red, nitrogen in blue, and sulphur in yellow, whilst the carbon atoms of the drug are depicted in green and the oxygens in red.

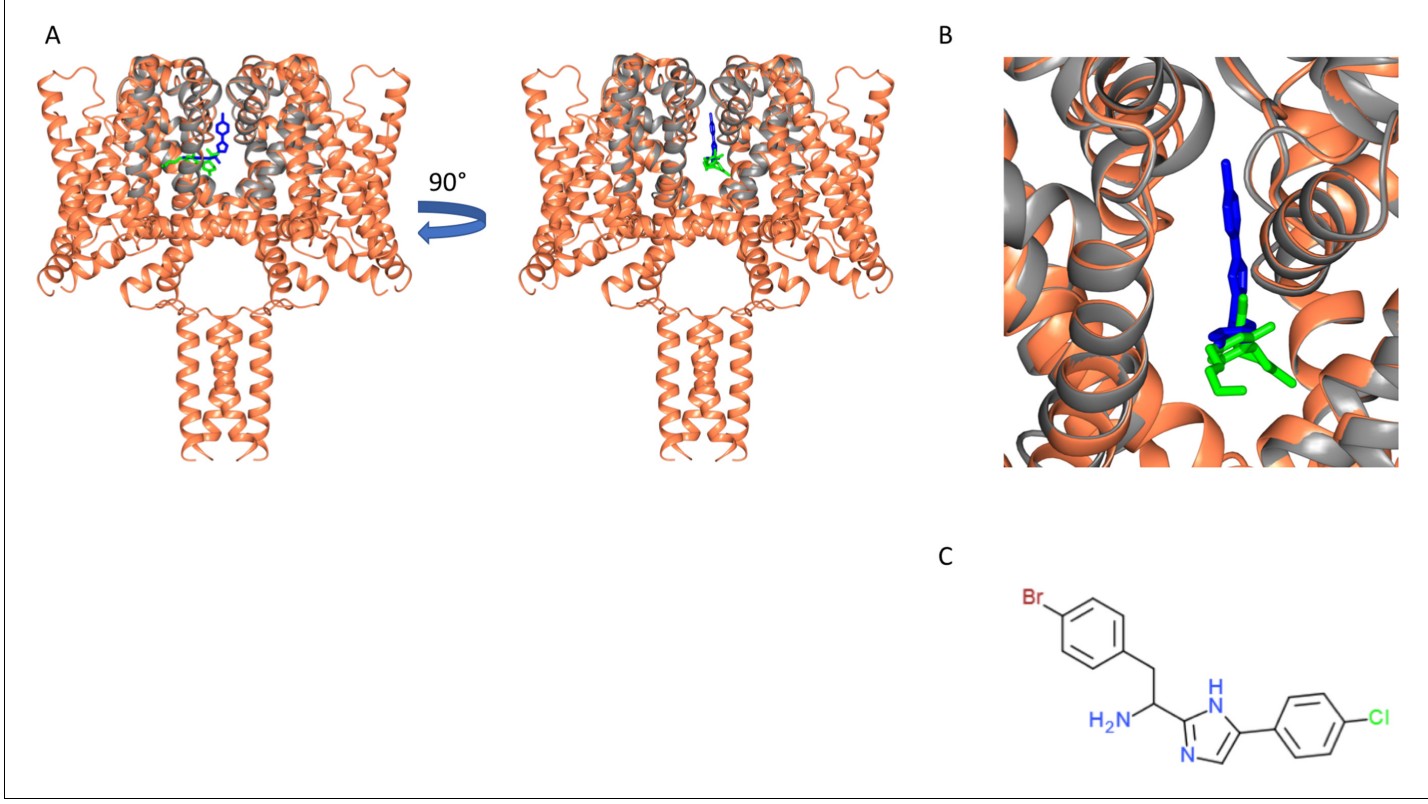

**Figure 4.** Similarity of CBD and analgesic compound binding sites. (**A**) Structural alignment of the NavMs-CBD complex (protein in coral ribbon depiction, CBD in green stick depiction), with the structure of the NavMs pore-PI1 complex (*Bagnéris et al., 2014*) [the protein is in grey ribbon depiction and the PI1 molecule is in blue stick depiction]. PI1 is a highly potent designed analgesic compound, which binds to and inhibits flux through the NavMs channel (*Bagnéris et al., 2014*). Two views of the aligned structural complexes, rotated by 90 degrees, are shown. They show the similarity, but not identity of the binding sites of the two ligands. (**B**) Detailed view showing the locations of these molecules in the pore/fenestration area. (**C**) Chemical structure of the PI1 compound.

## CBD inhibition of NavMs function

To investigate whether CBD functionally inhibits NavMs, whole-cell voltage-clamp studies of transiently transfected cells were performed (*Figure 5*). Previous studies (*Ghovanloo et al., 2018*) had shown that CBD imparts little selectivity in inhibiting various voltage-dependent sodium channels, including the bacterial sodium channel NaChBac. CBD inhibitions of Nav channels have steep Hill-slopes (~2) from both resting- and inactivated- states, indicating that relatively small magnitude differences in CBD potency are a product of the difference in slope. In the present study it was shown that CBD inhibits NavMs less potently and with a slightly shallower Hill slope than the other Nav channels previously studied (*Ghovanloo et al., 2018 Figure 5A*). The moderate variation in CBD inhibition potency between human Navs and NavMs is consistent with previous reports using other Nav blockers (*Bagnéris et al., 2014*). Overall, these results show that CBD inhibits NavMs similarly to other Nav channels and, thus supports the proposed interaction inside the pore depicted in the NavMs structure as being functionally relevant.

To gain further functional insight into the molecular interactions between CBD and the NavMs pore, CBD block was measured using the T207A mutant channel. This residue is located in the CBD-binding site (*Figure 1E*), with its side chain involved in hydrophobic interactions with the drug. This threonine has the closest sidechain (3.7 Å) to the CBD molecule. The mutation of the threonine side chain to a smaller alanine side chain results in a modest reduction in the CBD block (*Figure 5B,C*), hence correlating its location with a functional effect.

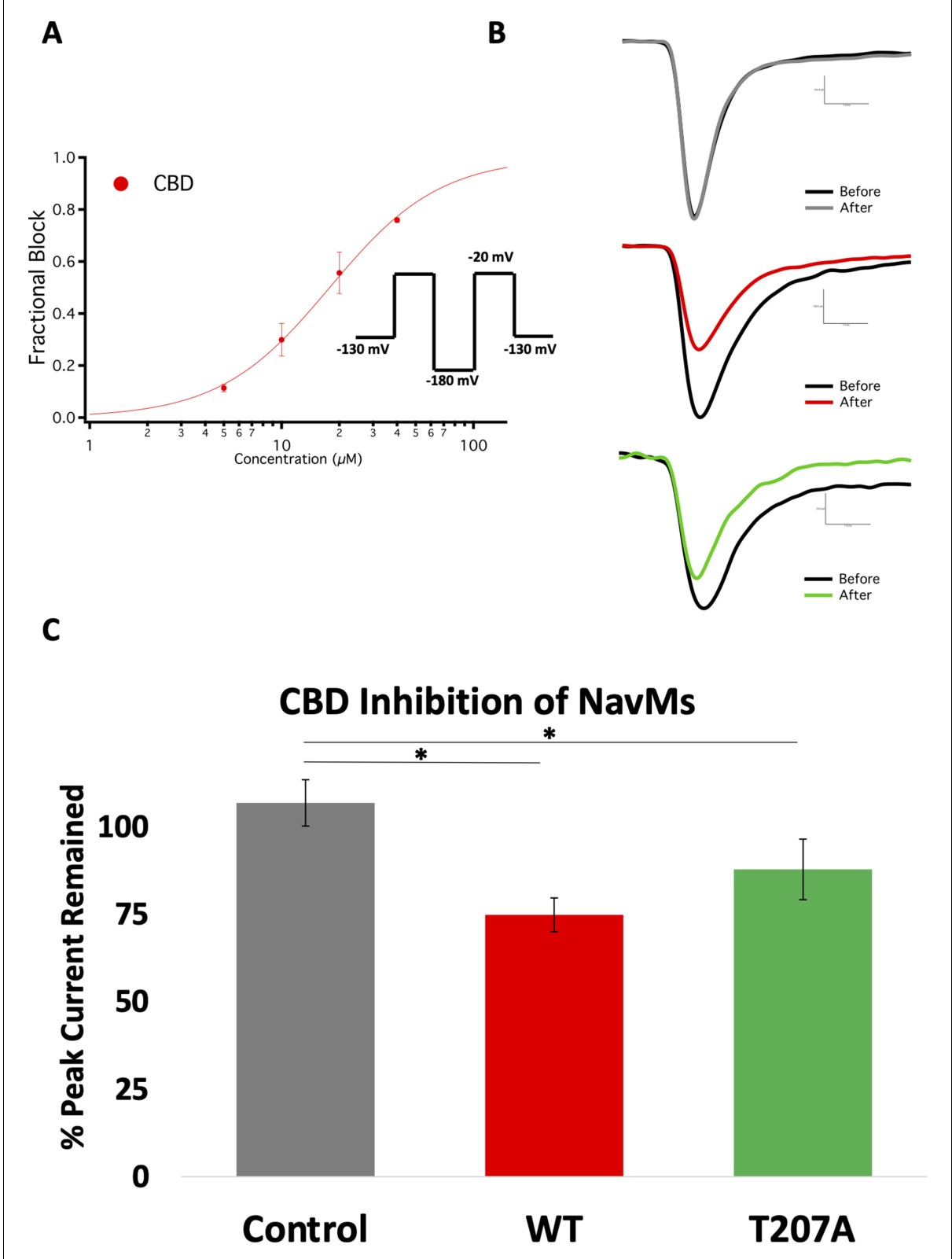

**Figure 5.** Electrophysiology studies of CBD inhibition of NavMs. (**A**) Block was measured after ~6 min wash and incubation in CBD. The $IC_{50}$ measurement was from CBD inhibition data obtained from whole-cell voltage-clamp recordings and fit with the Hill-Langmuir equation. The $IC_{50}$ for CBD inhibition of wild-type NavMs is $17.8 \pm 0.5$ μM with a Hill slope of $1.5 \pm 0.1$ (the S.E. values quoted are errors of the fit, n = 8 panel-wide). (**B**) Sample traces of (top) wild-type NavMs without CBD added, (middle) wild-type NavMs before (black) and after (red)10 μM CBD perfusion (compound

*Figure 5 continued on next page*

*Figure 5 continued*

effect was measured after ~6 min of wash/incubation, 1 Hz), and (bottom), as in the middle panel, but using the T207A mutant (in green). (C) Bar graphs showing percentage of peak sodium current remaining over time after control (no CBD) (grey) and with CBD (red, green, as above) perfusions at 10 µM (n = 4–6). Statistical comparison against control: WT (p<0.0027) and T207A (p=0.0269).

## Comparison of specificity/potential interactions with other hNavs

The focus of functional effects of CBD on hNavs has primarily been on hNav1.1, due to its association with epilepsy, although there is yet no structure for this isoform. However, due to the strong sequence similarities of hNav1.1, hNav1.2 and NavMs in the region identified as the binding site in NavMs (*Figure 6*), it has been possible to explore potential interactions using a hNav1.1 homology model based on the hNav1.2 cryo-EM structure (*Figure 6—figure supplement 1*). It suggests, not surprisingly, that the interactions would be very similar to those of Nav1.2 and NavMs in this region (*Figure 6—figure supplement 1*). In NavMs, the involvement of the T207 residue is of importance as it is well established as a primary binding site for local anaesthetics (and has been shown to be involved in the electrophysiology studies on NavMs reported herein). Furthermore, when the equivalent residue (F1774) was mutated in hNav1.1 (boxed in *Figure 6*), the binding affinity of CBD was found to decrease by a factor of ~2.5 (*Ghovanloo et al., 2018*). The binding site residues (coloured red in *Figure 6*) include both residues that are identical/homologous in NavMs and hNavs as well as residues that are only found in NavMs and not in human Navs. In most cases the non-cognate residues are also variable between hNavs and would thus appear not to be essential for the interactions.

**Figure 6.** Sequence alignments of CBD-binding site regions of NavMs With corresponding regions in hNav1.1 and hNav1.2. In red are the CBD-binding residues (within 3.9 Å of the compound) in NavMs and the equivalent residues in hNav1.1 and hNav1.2. The bold black F indicates the site of the NavMs F208L mutant used in these studies. It was changed from F to L in NavMs$_L$ because in half of the human Nav domains it is an F and in the other half it is an L (both indicated in bold black in the other sequences). However, as shown in *Figure 6—figure supplement 2*, the residue type present at this site makes essentially no difference in the structure. Residues located in the binding site are found within the P1 pore helix, the selectivity filter loop, and the S6 helix. The residue, which when mutated to alanine in hNav1.1 reduces the binding affinity of CBD (*Ghovanloo et al., 2018*), is boxed. This corresponds to T207 in NavMs, which is the residue that was mutated to alanine in the electrophysiology characterisations in the present study. The sequence alignment was carried out using Clustal Omega (*Sievers et al., 2011*) and annotated manually.

The online version of this article includes the following figure supplement(s) for figure 6:

**Figure supplement 1.** Alignment of the NavMs-CBD structure (coral) with a homology model of hNav1.1 (grey).

**Figure supplement 2.** Comparisons of wild type NavMs (gold) [PDB ID 5HVX] and the NavMs$_L$ mutant (coral) [PDB ID 6YZ2] structures.

It is notable, however, that the T207 residue that produced the altered electrophysiology results (*Figure 5*) is at a comparable site to the phenylalanine (F1774 in hNav1.1 – *Figure 6—figure supplement 1*) in the local anaesthetic site, which was also found to moderately alter the functon in the hNav1.1 orthologue (*Ghovanloo et al., 2018*).

## Modelling the binding of CBD relative to that of THC

There are two main phytocannabinoids that can be extracted from cannabis plants, the psychoactive tetrahydrocannabinol (THC) and the non-psychoactive CBD. Electrophysiological studies on hNavs and the bacterial NaChBac have identified CBD (*Ghovanloo et al., 2018*) as having functional effects that are distinct from those of THC on these channels. CBD and THC have previously been shown to inhibit hNav1.2 with similar potencies, however the THC inhibition slope was less steep than that of CBD (*Ghovanloo et al., 2018*).

The chemical structures of CBD and THC are very similar (*Figure 7A*), differing only by the presence of an additional free hydroxyl group on one of the rings in CBD (in THC the equivalent oxygen forms part of a closed pyran ring). Therefore, the structure of the CBD/NavMs complex was examined to see if it could provide a clue as to the reasons for the different functional effects of the two compounds. As can be seen in *Figure 7B*, by placing the THC structure into the CBD-binding site with the same orientation as found for CBD, it can be physically and sterically accommodated. However, and crucially, it is missing the one electrostatic interaction seen between CBD and NavMs: the hydrogen bond between the oxygen of the main chain residue M175 and the drug (*Figure 7C*). This is the consequence of the absence of the additional free hydroxyl group in THC, as noted above. That hydroxyl group is the one which forms the hydrogen bond present in the CBD-protein complex, and provides an additional intermolecular interaction for CBD, which could account for the differences in functional effects (inhibition slopes) of the two compounds on voltage-gated sodium channels.

## Comparison with binding to the TRPV2 channel

CBD has also been suggested to be a potential activator of the Transient Receptor Potential Cation Channel Subfamily V Member 2 (TRPV2) channel (*Qin et al., 2008*; *Morelli et al., 2013*), a channel which facilitates the non-specific movement of both sodium and calcium ions through plasma membranes. According to electrophysiology studies, CBD activates rat TRPV2 with an $EC_{50}$ of 3.7 μM (*Qin et al., 2008*), although the link with epilepsy (*Morelli et al., 2013*) is much less direct than that for sodium channels. Recently cryo-electron microscopy (cryo-EM) was used to elucidate the structure of TRPV2 in a CBD-bound state at a nominal resolution of 3.2 Å (*Pumroy et al., 2019*); that study indicated the presence of CBD in the pore region of the protein, thus supporting the proposal for TRPV2 being a candidate target for CBD binding. The general location of the CBD was visible in the structure, although the lower resolution of that structure did not allow detailed analysis of its binding site. However, its interactions appear to involve a number of hydrophobic side chains, whilst requiring a partial refolding of the adjacent region of the protein polypeptide. The binding site found for CBD in the TRPV2 structure is in a similar region to that of CBD in NavMs (*Figure 8*). However, the sodium channel-CBD site is located further into the fenestration than it is in TRPV2, but closer to the ion binding sites, and thus could more effectively modulate effects in the transmembrane passageway for ion conductance, and could account for sodium channels being blocked by CBD, whilst TRPV2 channels appear to be activated by them (*Morelli et al., 2013*).

## Discussion

This study has demonstrated the nature of the interactions of CBD and a voltage-gated sodium channel, showing that CBD-binding blocks the transmembrane pathway for sodium ion translocation through the membrane (*Naylor et al., 2016*), and hence provides a potential mechanism for the functioning of CBD in sodium channels. It further suggests a possible molecular basis for the medicinal effects of CBD in the treatment of epilepsies, as sodium channels have been shown to be causally-related to various types of human epilepsy, with disease-related mutations interfering with sodium ion transmembrane flux. The CBD-binding site is a novel site, near to, but not coincident with, known analgesic binding sites in sodium channels. The binding site is located at the pore end of the transmembrane fenestrations which enable the ingress of hydrophobic molecules into the channel lumen, hence this may also provide the pathway for CBD to enter and block the channels.

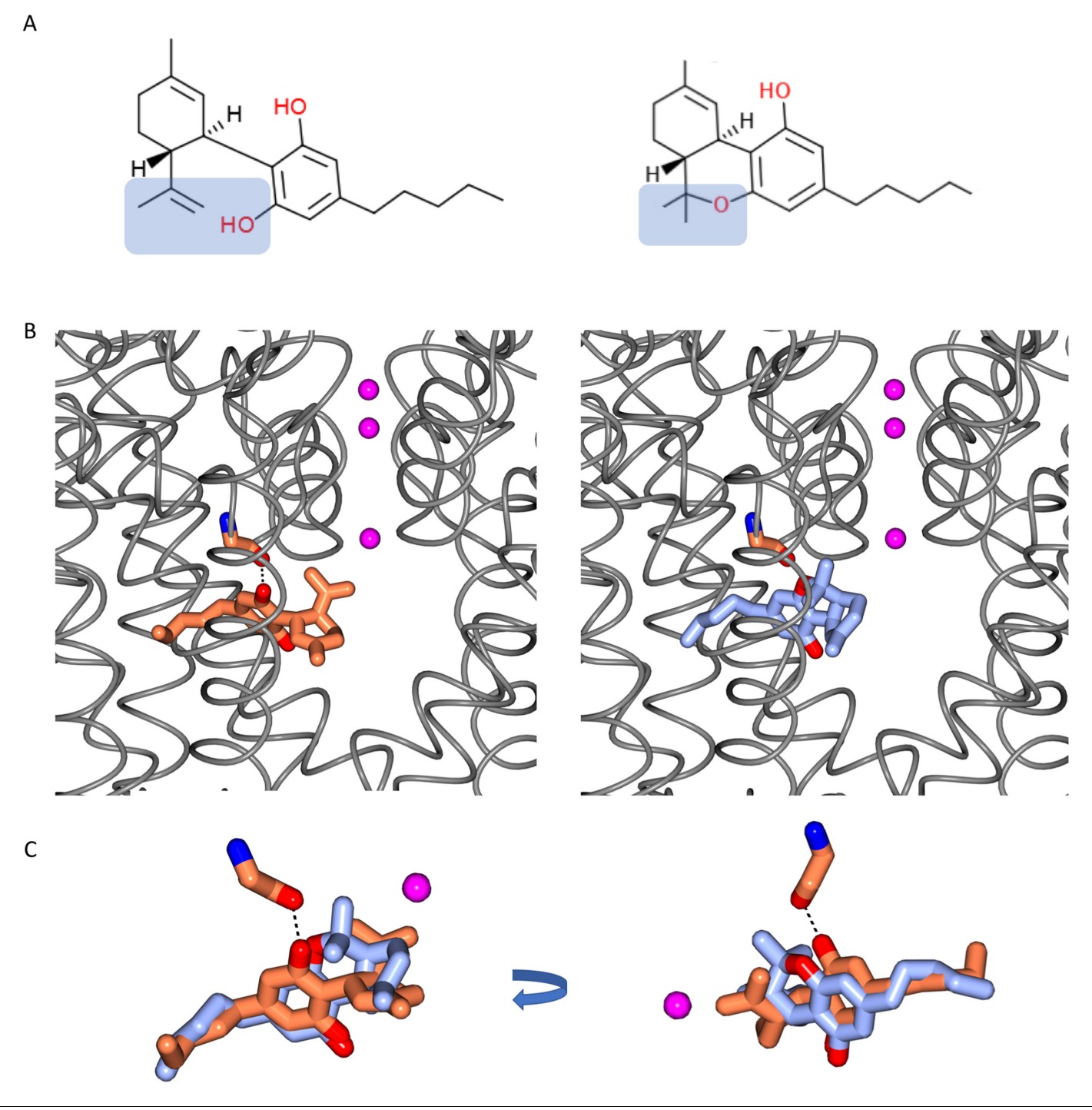

**Figure 7.** Comparisons of CBD and THC. (**A**) Chemical structures of cannabidiol (CBD) (left) and Δ-9-tetrahydrocannabinol (THC) (right). The difference between the two structures is highlighted in blue background. The formation of a pyran ring in THC removes the hydrogen from the hydroxyl group which is present in CBD. (**B**) The locations of CBD in the CBD-NavMs crystal structure (left), and THC (right) modelled into this site. The NavMs structure is depicted in grey ribbons, and the three sodium ions sites in the selectively filter are indicated by the pink balls, as a reference point. (**C**) (left) A detailed overlay of CBD (coral) and THC (blue) shows the additional hydrogen bond between the protein and drug for CBD, by comparison to THC, which, without the corresponding hydroxyl group does not have the potential to form such a hydrogen bond. The view on the right is rotated from the view on the left to clearly visualise the alignment.

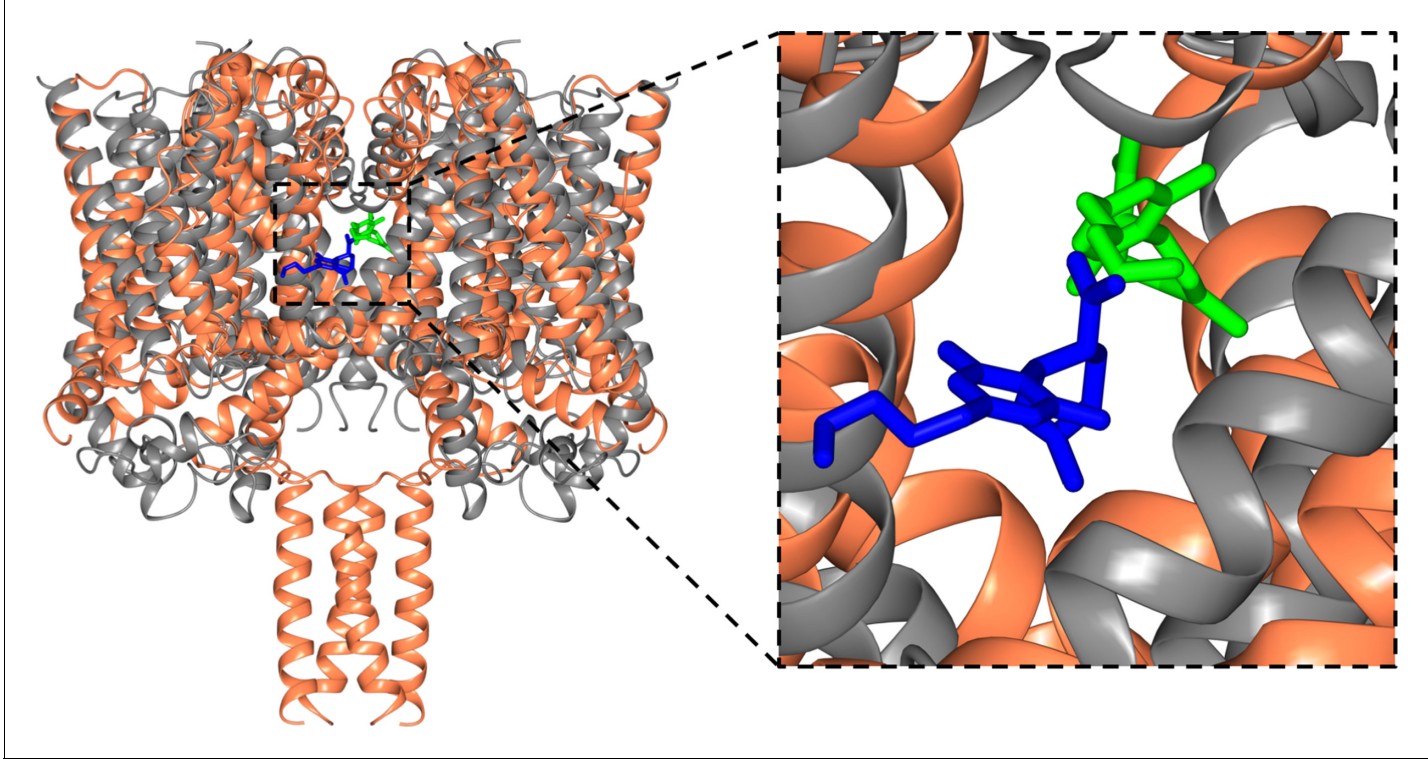

**Figure 8.** Structural alignments of the NavMs-CBD (coral ribbons) crystal structure and the TRPV2-CBD cryo-EM structure [PDB ID 6U88] (grey ribbons). (Left) Overall alignment of the structures. The TRPV2 structure was trimmed to remove the extramembranous regions for clarity. The RMSD of the alignment is 4.2 Å. The CBD in NavMs is shown in green and that in TRPV2 is in blue. The CBD site in NavMs appears to be located further into the fenestration than it is in TRPV2. (Right) Detailed view of the CBD sites, highlighting the similarity and differences in orientation and location in the two channel types.

Examination of the residues involved in the binding site interactions and modelling of the THC into the CBD-binding site have indicated a possible reason for why the closely-related psychoactive phytocannabinoid THC has not been observed to have a similar effect on sodium channel function: THC would be able to physically fit in the CBD site when oriented in the same manner, but it does not have the same hydroxyl moiety that in CBD forms an important hydrogen-bonding interaction with the channel protein.

A recent cryo-EM structural study (at lower resolution) has suggested that the TRPV2 channel may be a CBD-binding target, although that study did not show the relationship of the binding site to epilepsy-based mutations. Functional studies suggest these two channels act by different mechanisms: CBD is a channel blocker in NavMs, whilst in TRPV2 channels it appears to be a channel activator. In addition, NavMs channel and TRPV2 have quite different overall molecular folds: the NavMs channel is rather narrow, enabling sodium ion selectivity, whilst TRPV2 channel is able to act as a conduit for much larger substrates. Nevertheless, it is interesting that the binding sites for CBD in NavMs and TRPV2 (*Figure 8*) appear to be in a roughly comparable structural feature: near the transmembrane channel and substrates (sodium ions in the case of NavMs) pathways, and adjacent to the fenestration pathways that are proposed to enable drugs to enter into the channels.

In summary, this study has provided high-resolution structural evidence, along with functional studies, elucidating the molecular basis of the interactions of CBD, a drug recently approved for treatment of epilepsy, with a voltage-gated sodium channel target.

# Materials and methods

## Key resources table

| Reagent type (species) or resource | Designation | Source or reference | Identifiers | Additional information |
|---|---|---|---|---|
| Gene (Magnetococcus marinus MC-1) | NavMs | Uniprot | Mmc1_0789 | A0L5S6_MAGMM |
| Strain, strain background (Escherichia coli) | Over Express C41 (DE3) | Sigma-Aldrich | CMC0017 | Chemically competent E. coli for expression of toxic proteins |
| Cell line (Cricetulus griseus) | CHO-K1 | CedarLine Laboratories | RRID:CVCL_0214 | |
| Recombinant DNA reagent | pET15b plasmid encoding NavMs | PMID:28205548 | | Plasmid for NavMs and NavMsL expression for structural study |
| Recombinant DNA reagent | His-NavMs | addgene.org | 100004 | pTracer-CMV2, IRES GFP plasmid encoding NavMs |
| Sequence-based reagent | F208L_F | This Paper | PCR Primer for NavMsL (forward) | 5'-CTCACCACCCTGACCGTGCTCAACCTGTTTATTGG-3' |
| Squence-based reagent | F208L_R | This Paper | PCR Primer for NavMsL (reverse) | 5'-GAGCACGGTCAGGGTGGTGAGCATGATG+AACGGGATG-3' |
| Chemical compound, drug | Cannabidiol (CBD) | Sigma-Aldrich | C7515 | |
| Chemical compound, drug | Cannabidiol (CBD) | Toronto Research Chemicals | F175300 | |
| Software, algorithm | XDS | PMID:20124692 | RRID:SCR_015652 | Data Processing |
| Software, algorithm | Aimless | doi:10.1107/S0907444913000061 | RRID:SCR_015747 | Data Processing |
| Software, algorithm | CCP4 | PMID:15299374 | RRID:SCR_007255 | Structure Determination /Refinement |
| Software, algorithm | Phaser | PMID:19461840 | RRID:SCR_014219 | Structure Determination /Refinement |
| Software, algorithm | Coot | PMID:20383002 | RRID:SCR_014222 | Structure Determination /Refinement |
| Software, algorithm | REFMAC | PMID:21460454 | RRID:SCR_014225 | Structure Determination /Refinement |
| Software, algorithm | PROCHECK | doi:10.1107/s0021889892009944 | RRID:SCR_019043 | Structure Determination /Refinement |
| Software, algorithm | Molprobity | PMID:29067766 | RRID:SCR_014226 | Structure Determination /Refinement |
| Software, algorithm | BUSTER | PMID:22505257 | RRID:SCR_015653 | Structure Determination /Refinement |
| Software, algorithm | CCP4mg | PMID:21460457 | RRID:SCR_019041 | Structure Presentation |
| Software, algorithm | PatchMaster | HEKA Elektronik | RRID:SCR_000034 | Data Acquisition |
| Software, algorithm | FitMaster | HEKA Elektronik | RRID:SCR_016233 | Data Analysis |
| Software, algorithm | IGOR Pro | Wavemetrics, Lake Oswego, OR | RRID:SCR_000325 | Data Analysis |
| Software, algorithm | Clustal Omega | PMID:24170397 | RRID:SCR_001591 | Sequence Alignment |
| Software, algorithm | HOLE | PMID:9195488 | www.holeprogram.org | Channel Pore Dimension Analysis |

## Materials

Thrombin was purchased from Novagen Inc (Germany), decanoyl-N-hydroxyethylglucamide (Hega10) was purchased from Anatrace (USA); dimethyl sulfoxide (DMSO), sodium chloride, 2-amino-2-(hydroxymethyl)−1,3-propanediol (Tris), and imidazole were purchased from ThermoFisher Scientific (USA). Cannabidiol samples for structural and electrophysiology studies were purchased, respectively, from Sigma and Toronto Research Chemicals. Purification columns were purchased from GE Healthcare (USA). The F208L (NavMs$_L$) (*Figure 6—figure supplement 2*) mutation for crystallography was introduced by the SLIM site-directed mutagenesis protocol (*Chiu et al., 2004*), using the forward primer 5′-CTCACCACCCTGACCGTGCTCAACCTGTTTATTGG-3′ and reverse primer 5′-GAGCACGGTCAGGGTGGTGAGCATGATGAACGGGATG-3′. The sequence was verified by Source Bioscience, UK.

## Protein expression and purification

The NavMs (Uniprot ID A0L5S6) and NavMs$_L$ proteins were expressed and purified as previously described (*Sula et al., 2017*), with the following modifications: the bound protein was eluted in a buffer containing 20 mM Tris, pH 7.5, 300 mM NaCl, 0.5 M imidazole and 0.52% Hega10. The Histag was removed by thrombin cleavage overnight at 4° C. The protein sample was loaded onto a Superdex 200 column and eluted with 20 mM Tris, pH 7.5, 300 mM NaCl, and 0.52% Hega10 buffer. Protein samples were pooled and concentrated to 10 mg/ml using a 100 kDa cut-off Amicon concentrator and stored at a concentration of 10 mg/ml at −80°C.

## Crystallisation, data collection and structure determination

1 µl of cannabidiol (100 mM) in 100% DMSO was added to 50 µl of the purified protein solution to produce a final protein concentration of ~10 mg/ml containing 2 mM CBD and 2% v/v DMSO. The best crystals were grown at 4°C via the sitting drop vapour diffusion method using a 2:1 ratio of the protein and reservoir solutions containing 0.1 M sodium chloride, 0.1M lithium sulphate, 0.1 M HEPES, pH 7, and 30% v/v PEG300. The apo-NavMs$_L$ crystals were grown under the same condition as the crystals of the CBD complex, but without the DMSO and drug. Crystals were flash-frozen, with the PEG300 acting as the cryo-protectant. Data were collected on beamline P13 at the Electron Synchrotron (DESY, Germany); on beamline Proxima1 at the Soleil Synchrotron (France), and on beamlines IO3, IO4, and I24 at the Diamond Light Source (UK). Hundreds of crystals were screened and full data sets were collected from more than 40 crystals. Diffraction images were integrated and scaled using XDS (*Kabsch, 2010*) and then merged with Aimless (*Evans and Murshudov, 2013*) using the CCP4 suite of programmes (*Winn et al., 2011*). The structure was determined from the crystals which diffracted to the highest resolution (2.2 Å for the apo protein, and 2.3 Å for the CBD complex). Because of the small but significant variations in the unit cell dimensions and resolution between different crystals of the same type produced under the same conditions, as we have seen previously (*Naylor et al., 2016*; *Sula et al., 2017*), datasets from different crystals were not merged.

The structure determinations by molecular replacement were as previously described (*Sula et al., 2017*) using Phaser (*McCoy et al., 2007*) with the full-length wildtype NavMs structure (PDB 5HVX) as the search model. Model building was carried out using Coot (*Emsley et al., 2010*). Refinement was initially done using REFMACS (*Murshudov et al., 2011*), and then the refinement was continued using Buster (*Bricogne et al., 2019*). Data collection, processing and refinement statistics for both the apo and CBD complex structures are listed in *Supplementary file 1*. The structure quality was checked using PROCHECK (*Laskowski et al., 1993*) and MolProbity (*Chen et al., 2010*), which indicated that 100% of the residues were in allowed conformations. Figures were created in CCP4mg (*McNicholas et al., 2011*), unless otherwise noted.

## Electrophysiology

CBD dissolved in 100% DMSO was used to prepare extracellular solutions at different concentrations with no more than 0.5% total DMSO content. Chinese Hamster Ovary (CHOK1) cells were transiently co-transfected with cDNA encoding eGFP, the β1-subunit, and the NavMs α-subunit (https://www.addgene.org/100004/). Transfection was done according to the PolyFect transfection protocol. After

each set of transfections, a minimum of 8 hr incubation was allowed before plating on sterile cover-slips. All cells were incubated at 37°C/5% $CO_2$.

Whole-cell patch-clamp recordings were performed in an extracellular solution containing (in mM): 140 NaCl, 4 KCl, 2 $CaCl_2$, 1 $MgCl_2$, 10 HEPES (pH 7.4). Solutions were adjusted to pH7.4 with CsOH. Pipettes were filled with intracellular solution, containing (in mM): 120 CsF, 20 CsCl, 10 NaCl, 10 HEPES. All recordings were made using an EPC-9 patch-clamp amplifier (HEKA Elektronik, Lambrecht, Germany) digitized at 20 kHz via an ITC-16 interface (Instrutech, Great Neck, NY, USA). Voltage-clamping and data acquisition were controlled using PatchMaster software (HEKA Elektronik, Lambrecht, Germany) running on an Apple iMac. Current was low-pass-filtered at 10 kHz. Leak subtraction was performed automatically by software using a P/N procedure following the test pulse. Gigaohm seals were allowed to stabilize in the on-cell configuration for 1 min prior to establishing the whole-cell configuration. Series resistance was less than 5 MΩ for all recordings. Series resistance compensation up to 80% was used when necessary. All data were acquired at least 1 min after attaining the whole-cell configuration. Before each protocol, the membrane potential was hyperpolarized to −180 mV to ensure complete removal of inactivation. All experiments were conducted at $22 \pm 2$ °C. Analysis and graphing were done using FitMaster software (HEKA Elektronik) and Igor Pro (Wavemetrics, Lake Oswego, OR, USA). All data acquisition and analysis programs were run on an Apple iMac (Apple Computer).

Continuous variables are presented as means ± standard error and had a normal distribution. A T-test was used to compare the responses. A level of significance $\alpha = 0.05$ was used in all overall tests, and effects with p-values less than 0.05 were considered to be statistically significant.

## Additional information

### Funding

| Funder | Grant reference number | Author |
| --- | --- | --- |
| Biotechnology and Biological Sciences Research Council | BB/L006790 | BA Wallace |
| Biotechnology and Biological Sciences Research Council | BB/R001294 | BA Wallace |
| Rosetrees Trust | M848 | BA Wallace |
| Medical Research Council | Studentship | Lily Goodyer Sait |
| Natural Science and Engineering Research Council of Canada | RGPIN03920 | Peter C Ruben |
| Natural Science and Engineering Research Council of Canada | CGS-D:535333-2019 | Mohammad-Reza Ghovanloo |

The funders had no role in study design, data collection and interpretation, or the decision to submit the work for publication.

### Author contributions

Lily Goodyer Sait, Formal analysis, Funding acquisition, Investigation, Visualization, Writing - review and editing; Altin Sula, Conceptualization, Resources, Data curation, Formal analysis, Supervision, Validation, Investigation, Visualization, Methodology, Writing - original draft, Writing - review and editing; Mohammad-Reza Ghovanloo, Conceptualization, Formal analysis, Funding acquisition, Investigation, Methodology, Writing - review and editing; David Hollingworth, Conceptualization, Data curation, Formal analysis, Validation, Investigation, Methodology, Writing - original draft, Writing - review and editing; Peter C Ruben, Conceptualization, Funding acquisition, Methodology, Project administration; BA Wallace, Conceptualization, Funding Aquisition, Resources, Data curation, Formal analysis, Supervision, Validation, Investigation, Methodology, Writing - original draft, Writing - review and editing, Project Administration

## Author ORCIDs

Lily Goodyer Sait ![ORCID] https://orcid.org/0000-0003-4583-8790
Altin Sula ![ORCID] https://orcid.org/0000-0002-1820-4357
Mohammad-Reza Ghovanloo ![ORCID] https://orcid.org/0000-0002-2171-0744
David Hollingworth ![ORCID] https://orcid.org/0000-0001-7490-5310
Peter C Ruben ![ORCID] https://orcid.org/0000-0002-7877-5178
BA Wallace ![ORCID] https://orcid.org/0000-0001-9649-5092

## Decision letter and Author response

Decision letter https://doi.org/10.7554/eLife.58593.sa1
Author response https://doi.org/10.7554/eLife.58593.sa2

## Additional files

### Supplementary files

- Supplementary file 1. Crystal structure parameters.
- Transparent reporting form

### Data availability

Coordinates and Diffraction data have been deposited in the PDB under PDB6YZ2, and PDB6YZ0. All data generated or analysed during this study are included in the manuscript and supporting files.

The following datasets were generated:

| Author(s) | Year | Dataset title | Dataset URL | Database and Identifier |
|---|---|---|---|---|
| Sula A, Sait LG, Hollingworth D, Wallace BA | 2020 | Full Length Open-form Sodium Channel NavMs F208L | https://www.rcsb.org/structure/6YZ2 | RCSB Protein Data Bank, 6YZ2 |
| Sula A, Sait LG, Hollingworth D, Wallace BA | 2020 | Full Length Open-form Sodium Channel NavMs F208L in Complex with Cannabidiol (CBD) | https://www.rcsb.org/structure/6YZ0 | RCSB Protein Data Bank, 6YZ0 |

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
