## [Decision Letter]

**Acceptance summary:**

This manuscript presents high-quality X-ray diffraction and functional studies delineating the interaction of cannabidiol (CBD) with the prokaryotic voltage-gated sodium channel. These are important results that will help further our understanding of the effects of CBD on certain neurological diseases and also provide insight into the biophysics of sodium channels.

**Decision letter after peer review:**

Thank you for submitting your article "Cannabidiol Interactions with Voltage-Gated Sodium Channels" for consideration by *eLife*. Your article has been reviewed by three peer reviewers, including Leon D Islas as the Reviewing Editor and Reviewer #1, and the evaluation has been overseen by Olga Boudker as the Senior Editor.

The reviewers have discussed the reviews with one another and the Reviewing Editor has drafted this decision to help you prepare a revised submission.

Summary:

Cannabidiol (CBD) has been recently approved for treatment of epilepsy. Some of CBD anti-epileptic properties might be due to CBD inhibition of voltage-gated sodium channels but the molecular mechanism of such inhibition is unknown. Sait et al. studied the molecular bases of CBD inhibition using X-ray crystallography in application to the bacterial sodium channel NavMs. The authors solved NavMs structures in the apo state and in complex with CBD and based on structural comparison, identified CBD-binding sites and proposed the molecular mechanism of sodium channel inhibition by CBD. The crystal structures are of high quality and among the best published structures of sodium channels, and the study is without doubt of high importance. This is a solid manuscript from an experienced group that reports structural insights into cannabidiol interactions with the voltage-gated sodium channel NavM. The manuscript is easy to read, well-executed, and reveals interesting data.

Essential revisions:

The weakness of this study is the lack of functional data that would greatly complement the excellent structural results. Electrophysiological data showing the interaction of CBD with NavMs should be obtained and presented. This should be a very easy experiment to perform. CBD has been shown to block the NaChBac sodium channel, but there is no record in the literature that shows that CBD also blocks NavMs. This is a fundamental experiment that should be included in a revised version of the paper. It will also be of great interest to test the results of their structure by mutating appropriate sodium channel residues (e.g. in Nav1.1) and measure changes in cannabidiol interaction.

Similarly, discussion of the different ways CBD and THC bind to NavMs would greatly benefit from a comparison of the physiological effects of these two compounds. Does THC block NavMs and if it does, what is Kd/IC50 for THC compared to CBD? Electron density observed at the CBD site in the apo state structure needs to be shown side by side with the density for CBD in the structure obtained in the presence of CBD (a supplementary figure would suffice). Along these lines, it might be a good idea to add a brief discussion on how physiologically relevant is the apo state density. For example, if this site is always occupied by a lipid in physiological conditions, the channel would never open.

[Editors' note: further revisions were suggested prior to acceptance, as described below.]

Thank you for submitting your article "Cannabidiol Interactions with Voltage-Gated Sodium Channels" for consideration by *eLife*. Your article has been reviewed by the Reviewing Editor and Olga Boudker as the Senior Editor.

The reviewers have discussed the reviews with one another and the Reviewing Editor has drafted this decision to help you prepare a revised submission.

Summary:

This manuscript presents high quality X-ray diffraction studies delineating the interaction of cannabidiol (CBD) with the prokaryotic voltage-gated sodium channel. These are important results that will help further our understanding of the effects of CBD on certain neurological diseases and also provide insight into the biophysics of sodium channels.

Revisions:

The authors have done a superb job at addressing reviewers concerns and the manuscript is much stronger for that. In its present form, the number of principal figures is unnecessarily very large and reading the manuscript is a little "jumpy". Our request is that several figures should be combined into single figures with panels that present a more cohesive and consistent story.

---

## [Author Response]

Essential revisions:The weakness of this study is the lack of functional data that would greatly complement the excellent structural results. Electrophysiological data showing the interaction of CBD with NavMs should be obtained and presented. This should be a very easy experiment to perform. CBD has been shown to block the NaChBac sodium channel, but there is no record in the literature that shows that CBD also blocks NavMs.

We entirely agreed with the editor/reviewers that this was important information that should be included in the paper. Consequently we partnered with Prof. Peter Ruben’s expert electrophysiology group from Simon Fraser University, who have now undertaken and analysed the functional consequences of binding CBD to NavMs. The experiments are described in the Materials and methods, the results shown in the new Figure 12, and the interpretation of these discussed in the Results section. This is an important addition to the paper, and directly shows the functional effects that complement the structural studies.

This is a fundamental experiment that should be included in a revised version of the paper. It will also be of great interest to test the results of their structure by mutating appropriate sodium channel residues (e.g. in Nav1.1) and measure changes in cannabidiol interaction.

As requested, we mutated residue T207 (which corresponds with Nav1.1 residue F1774 [now indicated in the Figure 8 sequence and Figure 7 binding site figures] that showed reduced binding affinity for CBD), and also show the EP results for this mutant in new Figure 12. The NavMs T207A mutant protein does indeed show a diminished effect for CBD, indicating a reduced binding affinity. We thank the reviewer for this suggestion, as we believe this is an important addition to the manuscript.

Similarly, discussion of the different ways CBD and THC bind to NavMs would greatly benefit from a comparison of the physiological effects of these two compounds. Does THC block NavMs and if it does, what is Kd/IC50 for THC compared to CBD?

Because of drug licensing regulations, we have currently been unable to source highly purified THC for the experiments proposed by the reviewer. For that reason, we did molecular modelling/bioinformatics comparisons of the two compounds. Previously published studies in the literature (cited in this manuscript) comparing CBD and THC effects on hNav1.2 indicated very similar potencies, although the Hill slope for the THC was less steep, consistent with the possible missing electrostatic interaction in the NavMs binding site, as modelled. This is noted in the section on THC modelling.

Electron density observed at the CBD site in the apo state structure needs to be shown side by side with the density for CBD in the structure obtained in the presence of CBD (a supplementary figure would suffice).

As suggested, we have included a new figure (Figure 11), which clearly shows the requested comparisons of the electron density in the CBD-NavMs complex (Figure 11A) with that in apo-NavMs (Figure 11B), and includes a difference map (Figure 11C) for the CBD-NavMs complex with lipid placed in the CBD site. The CBD density is not in the same location nor is it the same shape as the density in the apo structure. The apo state density appears to be that of a single hydrocarbon chain of a lipid or detergent molecule (at partial occupancy), which adventitiously binds in an existing hydrophobic pocket. This hydrophobic region leads from the fenestration into the edge of the transmembrane channel region. The identity of this density as lipid is consistent with molecular dynamic studies of apo-NavMs (Ulmschneider et al., 2013) that have shown that lipid chains can enter and leave the fenestrations near this site during the course of a simulation; this is now noted in the “CBD-binding Site” section of the main text.

Along these lines, it might be a good idea to add a brief discussion on how physiologically relevant is the apo state density. For example, if this site is always occupied by a lipid in physiological conditions, the channel would never open.

Were it occupied by a lipid molecule all of the time (as suggested by the reviewer), for steric reasons it would block the entry of *any* hydrophobic drugs into the channel. Partial occupancy is consistent (see above) with the molecular dynamics studies (Ulmschneider et al., 2013) that have shown that lipid chains can enter and leave the fenestrations near this site during the course of a simulation. This is now mentioned in the Discussion section of the text.

[Editors' note: further revisions were suggested prior to acceptance, as described below.]

Revisions:The authors have done a superb job at addressing reviewers concerns and the manuscript is much stronger for that. In its present form, the number of principal figures is unnecessarily very large and reading the manuscript is a little "jumpy". Our request is that several figures should be combined into single figures with panels that present a more cohesive and consistent story.

We have followed your suggestions and changed the figures (decreasing the number of main figures from 14 to 8). This entailed merging several of them, deleting two that were unnecessary (previous Figures 2 and 3), and moving two of them to figure supplements, along with the associated changes to the figure legends and figure citations within the text. It did entail making some small changes to the organisation of the text, as also indicated in your email might be beneficial, without any loss/change of content. We believe this now reads much more coherently and are pleased to have done these changes.